# Elevated FIB-4 Is Associated with Higher Rates of Cardiovascular Disease and Extrahepatic Cancer History in Patients with Type 2 Diabetes Mellitus

**DOI:** 10.3390/biomedicines12040823

**Published:** 2024-04-09

**Authors:** Dimitrios S. Karagiannakis, Katerina Stefanaki, Foteini Petrea, Panagiota Zacharaki, Alexandra Giannou, Olympia Michalopoulou, Paraskevi Kazakou, Theodora Psaltopoulou, Vasiliki Vasileiou, Stavroula A. Paschou

**Affiliations:** 1Academic Department of Gastroenterology, Laiko General Hospital, School of Medicine, National and Kapodistrian University of Athens, 17 Agiou Thoma Street, 11527 Athens, Greece; 2Endocrine Unit and Diabetes Center, Department of Clinical Therapeutics, Alexandra Hospital, School of Medicine, National and Kapodistrian University of Athens, 11527 Athens, Greece; k.stefanaki@hotmail.gr (K.S.); zacharaki.p@hotmail.com (P.Z.); olympia1992@hotmail.com (O.M.); paraskevi.kazakou@gmail.com (P.K.); tpsaltop@hotmail.com (T.P.); s.a.paschou@gmail.com (S.A.P.); 3Department of Endocrinology, Alexandra Hospital, 11528 Athens, Greece; petrf055@gmail.com (F.P.); vasvasbee@gmail.com (V.V.)

**Keywords:** Type 2 diabetes mellitus, MASLD, liver fibrosis, FIB-4, cardiovascular disease, extrahepatic cancer

## Abstract

Background: Type 2 diabetes mellitus (T2DM) is often complicated by steatotic liver disease, cardiovascular disease (CVD), and extrahepatic cancer. We investigated whether FIB-4, an indicator of liver fibrosis, is associated with a higher risk of CVD and extrahepatic cancer history in T2DM. Methods: Two hundred and nine of 244 diabetics admitted to our center in one year were included and retrospectively evaluated. Results: One hundred and fifty-two (72.7%) were males and 57 (27.3%) females. The mean age and FIB-4 were 64.3 ± 11 years, and 1.15 ± 0.5, respectively. One hundred and fifty patients (71.8%) had FIB-4 ≤ 1.3, and 59 (28.2%) had FIB-4 > 1.3. A history of CVD was presented in 76 (36.4%) patients, and of extrahepatic cancer in 39 (18.7%). Patients with CVD were significantly older than those without (68.4 ± 8.5 vs. 63.2 ± 11.5 years; *p* = 0.002), with significantly higher FIB-4 (1.26 ± 0.5 vs. 1.08 ± 0.5; *p* = 0.012). Patients with cancer were older, with higher FIB-4 compared to those without (68.2 ± 9.5 vs. 64.4 ± 10.9 years; *p* = 0.098 and 1.37 ± 0.6 vs. 1.1 ± 0.5; *p* = 0.004, respectively). FIB-4 > 1.3 was associated with a 2.1-fold probability for CVD (χ^2^ = 5.810; *p* = 0.025) and 2.7-fold probability for cancer history (χ^2^ = 7.603; *p* = 0.01). Conclusions: FIB-4 ≥ 1.3 is associated with a higher probability of CVD or extrahepatic cancer history. FIB-4 could potentially discriminate patients at risk, justifying stricter surveillance.

## 1. Introduction

Type 2 diabetes mellitus (T2DM) is a significant medical condition that has become increasingly prevalent over the last few decades. It has been estimated that in the next 25 years, approximately 783 million people will be diagnosed with T2DM [1]. Moreover, T2DM mortality rates are expected to rise due to diabetes-related complications such as cardiovascular diseases (CVD), chronic kidney disease (CKD), retinopathy, and peripheral neuropathy [2].

Except for these well-recognized complications, T2DM is often associated with metabolic dysfunction-associated steatotic liver disease (MASLD), a major cause of liver disease worldwide, with a prevalence rate close to 30–35% in the general population and 60% in T2DM patients in particular [3]. Obesity, insulin resistance, metabolic syndrome, hyperlipidemia, hypertension, and T2DM comprise the precipitating factors of MASLD development [4]. The latter encompasses a spectrum of diseases ranging from simple hepatic steatosis through metabolic-associated steatohepatitis (MASH), defined by inflammation and hepatocellular necrosis, to liver fibrosis, cirrhosis, and hepatocellular carcinoma (HCC) [5]. Severe fibrosis is associated with an increased risk of liver-related morbidity and mortality in MASLD [6]. Furthermore, recent studies have shown a significant correlation between liver fibrosis and a higher risk of CVD and extrahepatic cancers [7,8,9,10]. A liver biopsy represents the gold standard method for estimating liver fibrosis. However, as an invasive procedure, it is associated with patient discomfort, sampling variability, and potential risk of complications [11,12]. Noninvasive elastographic methods and serum markers have successfully substituted liver biopsy in liver fibrosis assessment [13]. Among them, the fibrosis 4 index (FIB-4), an easy-to-perform, non-invasive test, has been proposed by the European Association for the Study of the Liver as a first-step screening tool to evaluate the presence of liver fibrosis. Specifically, FIB-4 ≤ 1.3 has been found to have a negative predictive value (NPV) >90% in ruling out significant liver fibrosis [13]. However, the universal use of FIB-4 in unselected populations would likely only reveal significant fibrosis in less than 5% of individuals worldwide with MASLD [14]. Hence, it would be more effective to use FIB-4 in specific high-risk groups, such as T2DM patients, who have a tenfold higher possibility of severe fibrosis than the general population [15]. In these subjects, an accurate evaluation of liver fibrosis is a matter of great importance, as a strong correlation between the degree of liver fibrosis and the possibility of T2DM-related complications independently of the glycated hemoglobin levels has been reported [16].

However, although the ability of FIB-4 ≤ 1.3 to effectively exclude liver fibrosis in patients with MASLD has been confirmed, its accuracy, particularly in T2DM, is disputed. Kim et al. reported low accuracy of FIB-4 for liver fibrosis detection in T2DM patients, while Gracen et al. found FIB-4 ≤ 1.3 to have a modest NPV (78.5%) in excluding significant fibrosis in T2DM [17,18]. In a recent study, Ajmera V et al. showed that an FIB-4 cut-off ≤1 might increase the method’s sensitivity in ruling out significant fibrosis in T2DM patients from 81% (for the threshold of 1.3) to 96%, minimizing false negative results [19]. Furthermore, Kawata et al. have elucidated that the use of a single FIB-4 cut-off (FIB-4 ≤ 1.3 vs. >1.3) in patients with T2DM might give misleading information regarding the possibility of liver-related complications’ development, such as hepatocellular carcinoma (HCC), as they found an elevating prevalence of HCC from 0% in FIB-4 < 1.3 to 0.8% in 1.3 ≤ FIB-4 < 2.67 and 14.3% in FIB-4 ≥ 2.67, respectively [20]. The authors also identified that an FIB-4 cut-off of 2.96 could detect the presence of underlying cirrhosis in patients with T2DM, with a sensitivity of 77%, a specificity of 94%, an NPV of 76%, a PPV of 94%, and a diagnostic accuracy of 84% [20].

In addition, there is a question about whether a universal cut-off value of FIB-4 should be used in all patients regardless of age or whether an adjusted-to-age FIB-4 might be more accurate. This question arises because age constitutes a component of the FIB-4 formula. Interestingly, according to Ishiba et al.’s multicenter analysis, higher FIB-4 thresholds could improve the diagnostic accuracy of FIB-4 in detecting advanced liver fibrosis in older patients [21]. A recent study involving 634 patients has revealed that the FIB-4 specificity for advanced fibrosis decreases with age, particularly in individuals aged 65 or above, where it drops to an unacceptably low rate of 35%. However, the study found that an FIB-4 cutoff of ≥2, when applied to patients over 65 years, can offer a higher specificity rate of 70% in diagnosing significant fibrosis without negatively impacting the sensitivity [22].

As the potential association between FIB-4 and the risk of complications has not been highly validated in T2DM patients, we conducted this study to address this issue. Thus, we investigated whether patients with T2DM-related complications, such as a history of CVD or a history of extrahepatic cancer, have higher FIB-4 levels compared to those without. Furthermore, we evaluated the feasibility of utilizing the FIB-4 cut-off value of 1.3 for all patients with T2DM, independent of age, to differentiate those with a higher susceptibility to a history of CVD or extrahepatic cancer. Additionally, we examined whether the FIB-4 threshold of 2 could be more precise in identifying patients over the age of 65 who are at risk for a history of CVD or extrahepatic cancer.

## 2. Material and Methods

### 2.1. Patients

Consecutive patients admitted to our diabetes center in one year (from 1 January 2023 up to 31 December 2023) were recruited. Inclusion criteria: age older than 18 years, a history of T2DM diagnosis for at least one year before the recruitment. Patients with liver disease of any etiology, such as chronic hepatitis B or C, autoimmune chronic liver disease, cholestatic liver disease, excessive alcohol consumption (>30 g/day for men and >20 g/day for women), hemochromatosis, Wilson disease, as well as active hypothyroidism, polycystic ovary syndrome, or any other endocrinopathies, and treatment with any hormonal medication or drugs with potential impact on metabolism or liver steatosis or fibrosis (apart from the antidiabetic treatment), were finally excluded. Two hundred and forty-four patients were admitted to our diabetic center during the above period. Of them, 35 patients were excluded from the study [4 due to active hypothyroidism; 8 due to a history of excessive alcohol consumption; 10 due to medical treatment with a potential impact on the liver; 3 due to chronic hepatitis B (positive surface antigen, HBsAg+); 6 due to a deficient medical history; and 4 due to denial to complete informed consent]. Finally, 209 patients were included in the study.

### 2.2. Methods

In all patients included, epidemiological characteristics (age, nationality, sex) and somatometric features (height, weight, BMI: body mass index) were recorded. Biochemical blood tests [HDL: high-density lipoprotein (kit EEA012, Thermofisher Scientific, Waltham, MA, USA); LDL: low-density lipoprotein (kit EEA014, Thermofisher Scientific, Waltham, MA, USA); Tg: triglycerides (kit EEA028, Thermofisher Scientific, Waltham, MA, USA); HbA1c: glycated hemoglobin A1 (kit 80099, Crystal Chem, Zaandam, The Netherlands)] were measured at the time of enrollment. All clinical events (CKD: chronic kidney disease; CD: coronary disease; HF: heart failure; stroke; extrahepatic cancer), as well as the history of hypertension, from the time of T2DM diagnosis to the time of recruitment were retrospectively evaluated. The specific antidiabetic treatment of patients at enrollment was also registered [Metformin; DPP-4: dipeptidyl peptidase-4; SGLT-2: Sodium-Glucose Transport Protein-2; GLP-1: glucagon-like peptide-1; Pioglitazone; Sulfonylureas; Insulin]. FIB-4 was calculated in all patients at enrollment using the following formula: Age (years) × AST (U/L)/[PLT(109/L) × ALT1/2 (U/L)] [23].

### 2.3. Ethics

The protocol was in accordance with the Helsinki Declaration for Human Studies, as revised in 1983 and approved by the institution’s ethics committee where the study was conducted. Written informed consent was obtained from all the participants for the anonymous use of their data.

### 2.4. Statistical Analysis

Statistical analysis was performed using SPSS software version 29 (SPSS Inc., Chicago, IL, USA). Continuous variables were tested for normality with the Kolmogorov–Smirnov test and described as the mean ± SD or median value [interquartile range (IQR)] as appropriate. Categorical variables were defined as the number of cases (percentages). Quantitative variables were compared between groups using the Student’s *t*-test or Mann–Whitney test for normally and non-normally distributed variables, respectively. Qualitative variables were compared using a corrected chi-square test or two-sided Fisher’s exact test, as appropriate. Binary logistic regression analysis was performed to identify independent factors associated with CVD and extrahepatic cancer history. *p* values of <0.05 were considered to be statistically significant.

## 3. Results

Two hundred and nine consecutive T2DM patients were retrospectively evaluated. The mean time from the diagnosis of diabetes to the enrollment was 9.83 ± 8.6 years (median: 8.2 years; IQR: 12 years). One hundred and fifty-two (72.7%) patients were male, and 57 (27.3%) were female. The mean age, BMI, glycated hemoglobin (HbA1c), and FIB-4 were 64.3 ± 11 years, 30.5 ± 5.8 kg/m^2^, 7.8 ± 2, and 1.15 ± 0.5, respectively. One hundred and fifty (71.8%) patients had FIB-4 ≤ 1.3, and 59 (28.2%) had FIB-4 > 1.3. Eighteen (8.6%) patients had an FIB-4 ≥ 2. Among the patients older than 65 years (*n* = 121), 14 (11.6%) had an FIB4 ≥ 2. Patients’ characteristics at enrollment are summarized in Table 1.

Seventy-six out of 209 (36.4%) patients had a history of CVD (CD/HF/stroke), and 39 (18.7%) had a history of extrahepatic cancer. Patients with CVD were significantly older (mean age: 68.4 ± 8.5 vs. 63.2 ± 11.5 years, respectively; *p* = 0.002), they were significantly heavier (93 ± 18 vs. 78 ± 20.7 kg; *p* = 0.018) and had significantly higher mean FIB-4 (1.26 ± 0.5 vs. 1.08 ± 0.5, respectively; *p* = 0.012) compared to patients without CVD. Furthermore, they had significantly lower LDL levels at the time of the enrollment (78.1 ± 34.6 vs. 103.4 ± 43.5; *p* < 0.001) and presented a history of hypertension and an FIB-4 > 1.3 in higher rates compared to patients without CVD (71.1% vs. 16.5%; *p* < 0.001 and 38.2% vs. 22.6%; *p* = 0.025, respectively). Patients with a cancer history were again older, having significantly higher mean FIB-4 at enrollment compared to those without a cancer history (mean age: 68.2 ± 9.5 vs. 64.4 ± 10.9 years; *p* = 0.098 and 1.37 ± 0.6 vs. 1.1 ± 0.5; *p* = 0.004, respectively). They also presented significantly more frequent FIB-4 > 1.3 compared to patients without cancer (46.1% vs. 24.1%; *p* = 0.01, respectively). Table 2 and Table 3 highlight the differences between patients with a history of CVD or extrahepatic cancer compared to those without.
—Differences between patients with FIB-4 > and ≤ 1.3 at the enrollment.

Between patients with FIB-4 >1.3 and ≤1.3, the former group was significantly older (70.2 ± 9.3 vs. 62.5 ± 10.9 years, *p* < 0.001) and had substantially lower HbA1c (7 ± 1.3 vs. 8 ± 2.1%; *p* < 0.001), lower Tg (157.9 ± 107.1 vs. 189.2 ± 131 mg/dL; *p* < 0.001), and lower LDL levels at the time of the enrollment (81 ± 5 vs. 100.7 ± 40.7 mg/dL; *p* = 0.003), compared to the latter group, respectively (Table 4).
—Correlation between FIB-4 > 1.3 in enrollment and a history of CVD.

A significant correlation was revealed between FIB-4 > 1.3 at enrollment and the presence of a CVD history (χ^2^ = 5.810, *p* = 0.025), as 103 out of 133 (77.4%) patients without CVD had FIB-4 ≤ 1.3, and 30 (22.6%) had FIB-4 > 1.3, while on the other hand, 47 (61.8%) of those with a CVD history had FIB-4 ≤ 1.3, and 29 (38.2%) had FIB-4 > 1.3 (Figure 1).

A significant relationship was found between FIB-4 > 1.3 at enrollment and a history of extrahepatic cancer (χ^2^ = 7.603, *p* = 0.01), as of the patients without cancer, 129 out of 170 (75.9%) had FIB-4 ≤ 1.3, and 41 (24.1%) had FIB-4 > 1.3. In contrast, in patients with a history of cancer (*n* = 39), 21 (53.8%) had FIB-4 ≤ 1.3, and 18 (46.2%) had FIB-4 > 1.3 at enrollment (Figure 2).

In the univariate analysis, age, sex, history of hypertension, diabetes duration, and FIB-4 > 1.3 at enrollment were associated with a history of CVD. In the multivariate analysis, sex (male: OR 0.266, 95%CI 0.114–0.619; *p* = 0.002), history of hypertension (OR 4.685, 95%CI 2.443–8.986; *p* < 0.001), and diabetes duration (OR 1.000, 95%CI 1.000–1.000; *p* = 0.02) were found to be independently associated with a history of CVD, while a tendency was revealed between FIB-4 > 1.3 at enrollment and a CVD history (*p* = 0.085) (Table 5).

In the univariate analysis, FIB-4 > 1.3 (OR 2.697, 95%CI 1.311–5.546; *p* = 0.007) and age (OR 1.036, 95%CI 1.000–1.073; *p* = 0.05) at enrollment were associated with a history of extrahepatic cancer. Due to FIB-4’s age-related nature, a multivariate analysis was not performed. Including FIB-4 and age in a multivariate analysis would likely result in questionable outcomes due to multicollinearity issues.
—Patients over 65 years old.

Among the 121 patients who were older than 65 years, 56 (46.3%) had a history of CVD. The FIB-4 score at enrollment did not significantly differ between those with a history of CVD and those without (1.35 ± 0.53 vs. 1.27 ± 0.58, respectively; *p* = 0.208). Of the 56 patients with a history of CVD, seven (12.5%) had an FIB-4 ≥ 2. No significant correlation was found between FIB-4 ≥ 2 at enrollment and a history of CVD (χ^2^ = 0.088, *p* = 0.784).

Twenty-four out of 56 (19.8%) patients older than 65 years had a history of extrahepatic cancer. Between those with and those without a history of extrahepatic cancer, no significant difference regarding the FIB-4 score at enrollment was identified (1.49 ± 0.65 vs. 1.26 ± 0.53, respectively, *p* = 0.087). Of the 24 patients with a history of extrahepatic cancer, four (16.7%) had FIB-4 ≥ 2 at the enrollment. No significant correlation between FIB4 ≥ 2 at enrollment and history of extrahepatic cancer was found (χ^2^ = 0.760, *p* = 0.474).

## 4. Discussion

T2DM is a severe disease often complicated by CKD, CVD, and cancer [2]. In addition, T2DM is commonly associated with MASLD, which represents the liver manifestation of metabolic syndrome. Nowadays, MASLD constitutes the most common liver disease worldwide and the leading cause of liver-related morbidity and mortality. Its prevalence has been rising in parallel with obesity, metabolic syndrome, and T2DM [24]. It displays a spectrum of diseases, ranging from simple steatosis to severe fibrosis and cirrhosis, with liver fibrosis having been shown to correlate not only with increased liver-related morbidity and mortality but also with increased risk of CVD and extrahepatic cancer development [4,7,25]. Therefore, it is essential to adequately detect and evaluate the presence and degree of liver fibrosis. Liver biopsy is the gold standard for assessing liver fibrosis, but its invasiveness makes it non-easy in clinical practice. Thus, several non-invasive methods, such as FIB-4, have been recommended as alternatives to liver biopsy to evaluate the presence of significant fibrosis in patients with MASLD [13,26]. Notably, an FIB-4 ≤ 1.3 has been shown to safely exclude the presence of significant fibrosis in patients with MASLD. However, there are concerns about the accuracy of this threshold in ruling out significant fibrosis in T2DM patients in particular [17,18,19]. Furthermore, it has not been verified whether this cut-off value is associated with an increased risk of CVD and extrahepatic cancers.

In this study, we showed that patients with a CVD history were older than those without, and more frequently had a history of hypertension, as well as significantly higher FIB-4 scores. In addition, a strong correlation was revealed between a value of FIB-4 over 1.3 at enrollment and a CVD history. There was no significant difference between CVD and non-CVD groups regarding LDL, Tg, and HDL values at enrollment. This is because patients with a history of CVD had already started taking hypolipidemic medication at the time of the event, and, therefore, they had an improved lipid profile when included in the study. A significant difference was observed at enrollment in the number of patients treated with GLP-1 analogs or SGLT-2 inhibitors between the CVD and the non-CVD group. This was mainly because a higher number of patients had started using the mentioned antidiabetic agents after developing CVD in order to benefit from their potential cardioprotective effects. Thus, patients with a history of CVD were already taking these pharmaceutical agents at higher rates at the time of inclusion in the study. Interestingly, patients with FIB-4 > 1.3 had significantly lower HbA1c, LDL, and Tg levels at enrollment than those with FIB-4 ≤ 1.3. This suggests that even an adequate regulation of sugar and lipid levels alone may not be enough to prevent liver fibrosis.

In the multivariate analysis, patients with hypertension were 4.7 times more likely to have a history of CVD. Additionally, females were found to be more prone to CVD as compared to men. Other investigators have already noticed the latter issue. Chadalavada et al., in a study of 22.685 diabetic patients from the UK Biobank study population, and Yoshida et al., in a cohort that was followed for up to 18.6 years, showed that female T2DM patients are at higher risk for CVD compared to males [27,28]. Moreover, a recent meta-analysis by Yaow et al. confirmed these findings by showing that females with T2DM have a higher risk than males of developing coronary heart disease (RR: 1.52, 95%CI: 1.32–1.76, *p* < 0.001), acute coronary syndrome (RR: 1.38, 95%CI: 1.25–1.52, *p* < 0.001), and heart failure (RR: 1.09, 95%CI: 1.05–1.13, *p* < 0.001) [29]. It has been speculated that women can accumulate more fat and experience a more significant insulin resistance, remaining in a prediabetic state for a more extended period compared to males before the development of T2DM. This prolonged stay in a prediabetic state may be responsible for the higher risk of CVD in female T2DM patients compared to males [29]. It should be noted that even though our study revealed that females with T2DM had a greater likelihood of having a history of CVD, the number of male participants who had a history of CVD was higher than that of females. The reason for that was probably the smaller number of female participants in our research (males/females: almost 3:1).

Regarding extrahepatic cancer, our study demonstrated that those patients with a history of cancer had significantly higher FIB-4 at enrollment and, more often, they had FIB-4 values over 1.3 compared to those without a history of cancer. Furthermore, a strong correlation between FIB-4 and cancer history was revealed, with a 2.7-fold higher probability of extrahepatic cancer history in patients with FIB-4 > 1.3 at enrollment. It is well-known that liver fibrosis and elevated FIB-4 are associated with an increased risk of hepatocellular carcinoma in patients with MASLD [30]. Moreover, very high levels of FIB-4 (>2.67) have been shown to correlate with an increased risk of colorectal cancer in MASLD [31]. However, to our knowledge, this is the first time that FIB-4 values over 1.3 were found to be associated with a history of extrahepatic cancer in patients with T2DM.

As FIB-4 is an age-dependent index, several studies have proposed different cut-offs according to patients’ age [17,18]. Thus, a threshold of ≥ 2 has been recommended in patients older than 65 [22]. In our study, patients of this age having such a high FIB-4 value were not more likely to have a history of CVD or extrahepatic cancer compared to patients with FIB-4 values lower than 2. However, drawing conclusive results on this matter is challenging due to our study’s limited number of patients over 65 with an FIB-4 score ≥2.

Our study has some limitations. First, the sample size is small enough to draw certain conclusions. Secondly, the retrospective evaluation of patients did not allow us to investigate the potential effect of the antidiabetic treatment on CVD and the risk of extrahepatic cancer. The antidiabetic treatment at enrollment was reported, but information about treatment modifications during T2DM and the duration of each administrated treatment is missing. Similarly, the data related to the lipid profiles of the patients were unfortunately not utilized to their full potential. The HDL, LDL, and Tg values were only recorded at the time of enrollment after patients had already commenced antilipidemic treatment. In contrast, data before the onset of CVD were not captured. In addition, sufficient data on FIB-4 at the time of diagnosis of T2DM were lacking. In our opinion, this does not downgrade the significance of our findings, as FIB-4 was not investigated as a predictor of future CVD events or cancer but as a marker associated with CVD or cancer history. Another limitation is the underrepresentation of females. As already mentioned, there were 3-fold more males than females. However, other studies have also observed gender disparity in T2DM representation. It seems that T2DM is more common in men than women, and in addition, women are diagnosed at older ages than men [32]. Thus, a higher representation of males is possible in a study that includes consecutive and no pre-selected T2DM patients, such as ours.

## 5. Conclusions

Diabetic patients with higher FIB-4 scores are more likely to have a history of CVD or extrahepatic cancer. There is a significant association between FIB-4 > 1.3 and the likelihood of having a history of CVD or extrahepatic cancer. Prospective studies involving larger cohorts, however, are necessary to investigate whether FIB-4 scores greater than 1.3 could be used as a predictor of CVD and cancer in patients with T2DM.

## Figures and Tables

**Figure 1 biomedicines-12-00823-f001:**
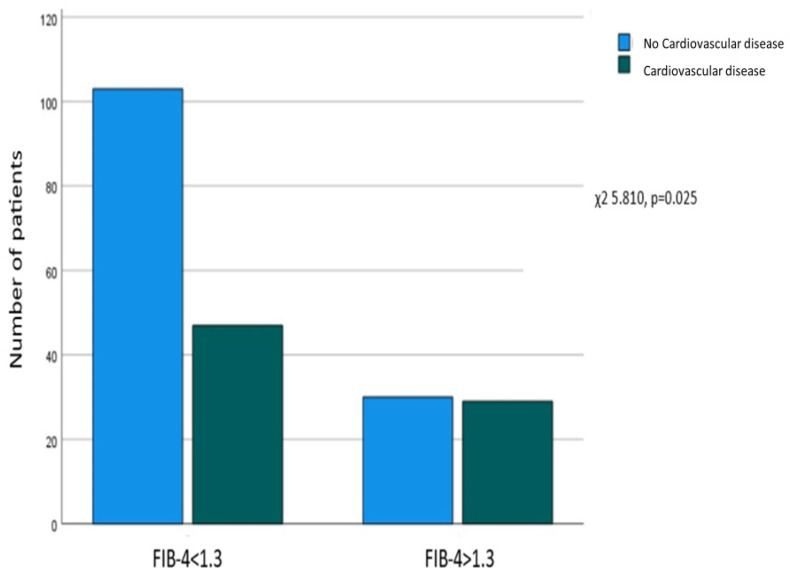
Association between FIB-4 at enrollment and a history of cardiovascular disease. Correlation between FIB-4 > 1.3 at enrollment and a history of extrahepatic cancer.

**Figure 2 biomedicines-12-00823-f002:**
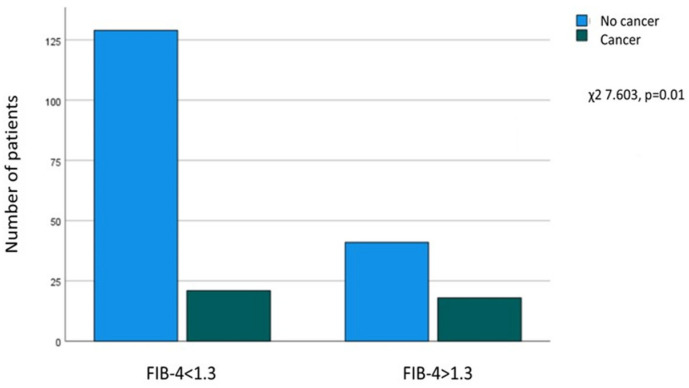
Association between FIB-4 at enrollment and a history of extrahepatic cancer.

**Table 1 biomedicines-12-00823-t001:** Patients’ characteristics at enrollment.

Sex (M/F)	152/57 (72.7%/27.3%)
Nationality (Greek/Other)	187/22 (89.5%/10.5%)
Age (years)	67 (15)
Weight (kg)	85 (110)
BMI (kg/m^2^)	29.2 98.4)
HbA1c (%)	7.2 (2.6)
LDL (mg/dL)	89 (61)
HDL (mg/dL)	43.5 (16)
Tg (mg/dL)	147 (100)
FIB-4	1.02 (0.5)
Metformin (Yes/No)	153/56 (73.2%/26.8%)
DPP-4 inhibitors (Yes/No)	47/162 (22.5%/77.5%)
SGLT-2 inhibitors (Yes/No)	64/145 (30.6%/69.4%)
GLP-1 analogs (Yes/No)	69/140 (33%/67%)
Sulfonylureas (Yes/No)	22/187 (10.5%/89.5%)
Pioglitazone (Yes/No)	7/202 (3.3%/96.7%)
Insulin (Yes/No)	79/130 (37.8%/62.2%)
Hypertension (Yes/No)	96/113 (45.9%/54.1%)
CKD (Yes/No)	14/195 (6.7%/93.3%)
CVD (Yes/No)	76/133 (36.4%/63.6%)
Extrahepatic cancer (Yes/No)	39/160 (18.7%/81.3%)
FIB-4 > 1.3 (Yes/No)	59/150 (28.2%/71.8%)

Variables are expressed as median value and interquartile range (IQR) or number of cases (percentages). BMI: body mass index; HbA1c: glycated hemoglobulin A1; LDL: low-density lipoprotein; HDL: high-density lipoprotein; Tg: triglycerides; DPP-4: dipeptidyl peptidase-4; SGLT-2: Sodium-Glucose Transport Protein-2; GLP-1: glucagon-like peptide-1; CKD: chronic kidney disease; CVD: cardiovascular disease. The table displays patients’ lipid values and antidiabetic treatment at the study's enrollment time.

**Table 2 biomedicines-12-00823-t002:** Differences at the time of enrollment between patients with and without a history of CVD.

Variables	Patients with CVD (*n* = 76, 36.4%)	Patients without CVD (*n* = 133, 63.6%)	*p* Value
Sex (M/F)	67/9 (88.2%/11.8%)	85/48 (63.9%/36.1%)	<0.001
Age (years)	69 (12) 68.4 ± 8.5	63 (16) 63.2 ± 11.5	0.002
Weight (Kg)	90 (25) 93 ± 18	84 (26) 78 ± 20.7	0.018
BMI (Kg/m^2^)	29.4 (8.4) 30.7 ± 5.4	28.9 (8.3) 30.4 ± 6.1	0.475
HbA1c (%)	7.5 (38) 7.6 ± 1.6	7 (2.8) 7.8 ± 2.1	0.938
LDL (mg/dL)	76 (38) 78.1 ± 34.6	105 (56) 103.4 ± 43.5	<0.001
HDL (mg/dL)	42 (15) 42.9 ± 12.1	44 (15) 44.8 ± 10.3	0.125
Tg (mg/dL)	120 (89) 161.1 ± 103.4	158 (106) 182.7 ± 127.9	0.215
FIB-4	1.1 (0.67) 1.26 ± 0.54	0.97 (0.49) 1.08 ± 0.5	0.012
Metformin	48 (63.2%)	105 (77.2%)	0.015
DPP-4 inhibitors	17 (22.4%)	30 (22.6%)	0.975
SGLT-2 inhibitors	31 (40.8%)	33 (24.8%)	0.019
GLP-1 analogs	33 (43.4%)	36 (27.1%)	0.021
Sulfonylureas	7 (9.2%)	15 (11.3%)	0.815
Pioglitazone	4 (5.3%)	3 (2.3%)	0.259
Insulin	30 (39.5%)	49 (36.8%)	0.767
Hypertension	54 (71.1%)	22 (16.5%)	<0.001
FIB-4 > 1.3	29 (38.2%)	30 (22.6%)	0.025

Variables are expressed as median value and interquartile range (IQR) or number of cases (percentages). (Mean value ± standard deviation (SD) is presented underneath.) M/F: male/female; BMI: body mass index; HbA1c: glycated hemoglobulin A1; LDL: low-density lipoprotein; HDL: high-density lipoprotein; Tg: triglycerides; DPP-4: dipeptidyl peptidase-4; SGLT-2: Sodium-Glucose Transport Protein-2; GLP-1: glucagon-like peptide-1; CVD: cardiovascular disease. The table displays patients’ lipid values and antidiabetic treatment at the study's enrollment time.

**Table 3 biomedicines-12-00823-t003:** Differences at the time of enrollment between patients with and without a history of extrahepatic cancer.

Variables	Patients with Extrahepatic Cancer (*n* = 39, 18.7%)	Patients without Extrahepatic Cancer (*n* = 170, 81.3%)	*p* Value
Sex (M/F)	28/11 (71.8%/28.2%)	124/46 (72.9%/27.1%)	0.845
Age (years)	67.5 (15) 68.2 ± 9.5	66 (14) 64.4 ± 10.9	0.098
Weight (Kg)	85 (13) 84.4 ± 14.9	86.5 (30) 90.9 ± 20.7	0.206
BMI (Kg/m^2^)	27.3 (6.4) 29 ± 4.7	29.7 (8.5) 30.8 ± 6	0.152
HbA1c (%)	7.2 (2.6) 7.5 ± 1.5	7.2 (2.6) 7.8 ± 2	0.563
LDL (mg/dL)	86 (47) 83.5 ± 31.7	91 (62) 96.7 ± 44	0.13
HDL (mg/dL)	47 (16) 45.8 ± 10.9	42.5 (14) 43.7 ± 11	0.2
Tg (mg/dL)	119.5 (101) 157.9 ± 107.1	150 (97) 178.8 ± 122.5	0.114
FIB-4	1.13 (0.64) 1.37 ± 0.6	0.96 (0.5) 1.1 ± 0.5	0.004
Metformin	27 (69.2%)	126 (74.1%)	0.551
DPP-4 inhibitors	10 (25.6%)	37 (21.7%)	0.671
SGLT-2 inhibitors	14 (35.9%)	50 (29.4%)	0.445
GLP-1 analogs	9 (23.1%)	60 (35.3%)	0.186
Sulfonylureas	4 (10.3%)	18 (10.6%)	0.951
Pioglitazone	2 (5.1%)	5 (2.9%)	0.617
Insulin	18 (46.1%)	61 (35.9%)	0.273
Hypertension	19 (61.5%)	77 (45.3%)	0.725
FIB-4 > 1.3	18 (46.1%)	41 (24.1%)	0.01

Variables are expressed as median value and interquartile range (IQR) or number of cases (percentages). (Mean value ± standard deviation (SD) is presented underneath.) M/F: male/female; BMI: body mass index; HbA1c: glycated hemoglobulin A1; LDL: low-density lipoprotein; HDL: high-density lipoprotein; Tg: triglycerides; DPP-4: dipeptidyl peptidase-4; SGLT-2: Sodium-Glucose Transport Protein-2; GLP-1: glucagon-like peptide-1. The table displays patients’ lipid values and antidiabetic treatment at the study's enrollment time.

**Table 4 biomedicines-12-00823-t004:** Differences in enrollment between patients with FIB-4 > 1.3 and FIB-4 ≤ 1.3.

Variables	Patients with FIB-4 > 1.3 (*n* = 59, 28.2%)	Patients with FIB-4 ≤ 1.3 (*n* = 150, 71.8%)	*p* Value
Sex (M/F)	49/10 (83%/17%)	103/47 (68.7%/31.3%)	0.039
Age (years)	71 (14) 70.2 ± 9.3	64 (14) 62.5 ± 10.9	<0.001
Weight (Kg)	87 (16.5) 88.6 ± 13	85 (30) 90.5 ± 21.5	0.712
BMI (Kg/m^2^)	28.4 (8.60 30 ± 4.9	29.4 (8.4) 30.7 ± 6.1	0.683
HbA1c (%)	6.6 (1.6) 7 ± 1.3	7.4 (2.5) 8 ± 2.1	<0.001
LDL (mg/dL)	71.5 (44.3) 81 ± 5	97 (55) 100.7 ± 40.7	0.003
HDL (mg/dL)	45 (14) 45.6 ± 10.3	43 (16) 44.2 ± 11.3	0.402
Tg (mg/dL)	119.5 (101) 157.9 ± 107.1	158 (102) 189.2 ± 131	<0.001
Metformin	33 (55.9%)	120 (80%)	<0.001
DPP-4 inhibitors	16 (27.1%)	31 (20.7%)	0.358
SGLT-2 inhibitors	25 (42.4%)	39 (26%)	0.03
GLP-1 analogs	18 (30.5%)	51 (34%)	0.744
Sulfonylureas	6 (10.2%)	16 (10.7%)	0.916
Pioglitazone	1 (1.7%)	6 (4%)	0.676
Insulin	24 (40.7%)	55 (36.7%)	0.636
Hypertension	31 (52.5%)	65 (43.3%)	0.281

Variables are expressed as median value and interquartile range (IQR) or number of cases (percentages). (Mean value ± standard deviation (SD) is presented underneath.) M/F: male/female; BMI: body mass index; HbA1c: glycated hemoglobulin A1; LDL: low-density lipoprotein; HDL: high-density lipoprotein; Tg: triglycerides; DPP-4: dipeptidyl peptidase-4; SGLT-2: Sodium-Glucose Transport Protein-2; GLP-1: glucagon-like peptide-1. The table displays patients’ lipid values and antidiabetic treatment at the study's enrollment time.

**Table 5 biomedicines-12-00823-t005:** Factors independently associated with a CVD history.

Univariate Analysis				Multivariate Analysis		
Variables	OR	95%CI	*p* Value	OR	95%CI	*p* Value
Age	1.051	1.020–1.082	<0.001			
Sex (male)	0.238	0.109–0.519	<0.001	0.266	0.114–0.619	0.002
Duration of diabetes	1.000	1.000–1.000	0.001	1.000	1.000–1.000	0.02
Hypertension history	5.318	2.872–9.846	<0.001	4.685	2.443–8.986	<0.001
FIB-4 > 1.3	2.118	1.144–3.923	0.017	1.835	0.920–3.659	0.085

CVD: cardiovascular disease; OR: odds ratio; CI: confidence interval. Although age was significantly associated with a history of CVD in the univariate analysis, it was not included in the multivariate one. This was done to avoid multicollinearity issues, as FIB-4 is per se an age-dependent factor: [Age (years) × AST (U/L)/[PLT(109/L) × ALT1/2 (U/L)] [23].

## Data Availability

The datasets generated and analyzed during the current study are available from the corresponding author upon reasonable request.

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
