# Peer review of "Elevated FIB-4 Is Associated with Higher Rates of Cardiovascular Disease and Extrahepatic Cancer History in Patients with Type 2 Diabetes Mellitus"

_biomedicines, 2024, doi:10.3390/biomedicines12040823_

Round 1
Reviewer 1 Report
Comments and Suggestions for Authors
As a preliminary step, in order to make their manuscript more readable, authors must provide for a thorough revision, including the tables.
I have some difficulty interpreting some of the results presented, such as the very high risk in women for CVD or the absence of association with lipid profile. The authors need to explain these findings.
Several evidences have demonstrated the anti-tumor effect of Metformin; it would have been useful for the Authors to correct analysis also for pharmacological treatment.
I think the major limitation is represented by the fact that the authors wanted to investigate in the same population the association of FIB-4 with both the risk of cardiovascular events and extrahepatic cancer. The sample size is modest enough to be able to reach certain conclusions.
In view of the proliferative effects of insulin in subjects with insulin resistance, it would have been useful to know the insulinemia values.
Comments on the Quality of English LanguageModerate editing of English language is required
Author Response
As a preliminary step, in order to make their manuscript more readable, authors must provide for a thorough revision, including the tables.
REPLY: We thank the reviewer for their comment. We carefully revised the entire manuscript and tables, correcting grammatical errors, rephrasing some sentences, and more thoughtfully describing our results. Furthermore, we corrected some mistakes in the tables. Hence, no more differences exist between the tables' results and those provided in the manuscript's main body.
I have some difficulty interpreting some of the results presented, such as the very high risk in women for CVD or the absence of association with lipid profile. The authors need to explain these findings.
REPLY: Regarding the high risk of CVD in women with T2DM, this has already been reported by other investigators as well (references 26,27,28). It has been speculated that women can accumulate more fat and experience a more significant insulin resistance, remaining in a prediabetic state for a more extended period compared to males until the development of T2DM. This prolonged stay in a prediabetic state may be responsible for the higher risk of CVD in female T2DM patients compared to males. We have mentioned that in the discussion (page 16, 2nd paragraph).
Regarding the absence of any association between CVD and the lipidemic profile, we have mentioned that the lipid profile provided is at the time of enrollment. Due to the study's retrospective nature, the HDL, LDL, and Tg values before the onset of CVD are lacking. Unfortunately, patients with a history of CVD had already started taking hypolipidemic medication at the time of the event, and, therefore, they had an improved lipid profile when included in the study. That is why an association between the lipid profile at enrollment and a CVD history was not found. We clarified that in the discussion (page 15, last paragraph, and on page 17, 3rd paragraph, where the study's limitations are highlighted).
Several evidences have demonstrated the anti-tumor effect of Metformin; it would have been useful for the Authors to correct analysis also for pharmacological treatment.
REPLY: We did not have information about patients' medical treatment during the whole course of T2DM. We had information about antidiabetic therapy at enrollment. We have mentioned that limitation on page 15, last paragraph, and on page 16, 1st paragraph ("A significant difference was observed at enrollment in the number of patients treated with GLP-1 analogs or SGLT-2 inhibitors between the CVD and the non-CVD group. This was mainly because a higher number of patients had started using the mentioned antidiabetic agents after developing CVD in order to benefit from their potential cardioprotective effects. Thus, patients with a history of CVD were already taking these pharmaceutical agents at higher rates at the time of inclusion in the study."). We have also mentioned it on page 17, 3rd paragraph (" the retrospective evaluation of patients did not allow us to investigate the potential effect of the antidiabetic treatment on CVD and the risk of extrahepatic cancer. The antidiabetic treatment at the enrollment was reported, but information about treatment modifications during T2DM and the duration of each administrated treatment is missing".)
Nevertheless, according to the reviewer's proposal, we analyzed whether patients taking Metformin at the time of recruitment had lower rates of extrahepatic cancer history. No significant association was found between metformin treatment and cancer history (χ2 0.386, p=0.551).
I think the major limitation is represented by the fact that the authors wanted to investigate in the same population the association of FIB-4 with both the risk of cardiovascular events and extrahepatic cancer. The sample size is modest enough to be able to reach certain conclusions.
REPLY: We believe that the reviewer is right. It is difficult to draw certain conclusions as our study's sample size is small. We mentioned that in the discussion (study's limitations, page 17, 3rd paragraph). We also changed the conclusions to clarify that future prospective studies with larger populations are necessary (page 18, 1st paragraph). However, we believe that our study has some strengths which make it valuable enough. First, it was confirmed that T2DM patients with a CVD or cancer history have higher FIB-4 scores compared to those without a CVD or cancer history. More importantly, it is the first time that the FIB-4 value of 1.3 was shown to be correlated with higher rates of CVD and cancer history. We believe that our results will urge investigators to conduct further studies on this issue. If future studies confirm these findings, FIB-4>1.3 could differentiate patients at higher risk of CVD and cancer regardless of age.
In view of the proliferative effects of insulin in subjects with insulin resistance, it would have been useful to know the insulinemia values.
REPLY: Unfortunately, due to the study's retrospective nature, information about insulin values is lacking. Comments on the Quality of English LanguageModerate editing of English language is required
REPLY: Editing the English language was performed by a colleague living and working in the UK.
Reviewer 2 Report
Comments and Suggestions for Authors
Excellent work!This is a very good article, with a topic of great interest regarding hepatic steatosis in patients with diabetes and the way in which these patients can be evaluated, monitored and diagnosed.
The article is well structured, without significant mistakes, with an appropriate and up-to-date bibliography.
Author Response
Excellent work!This is a very good article, with a topic of great interest regarding hepatic steatosis in patients with diabetes and the way in which these patients can be evaluated, monitored and diagnosed.
The article is well structured, without significant mistakes, with an appropriate and up-to-date bibliography.
REPLY: We thank the reviewer for their kind words.
Reviewer 3 Report
Comments and Suggestions for Authors
I think the study can be relevant; however, your results do not match your conclusion- the Fib 4 is not associated with higher risk of CVD or extrahepatic cancer. Fib 4 is not statistically significant for CVD in the multivariate analysis and the results for extrahepatic cancer were missing. Although as you note that age is a component of FIb-4, you removed age from the model and Fib-4 became not significant, this suggests that age may be a significant contributor to the observed relationship. In fact, given the age of the population of study, a better Fib-4 cut off to use may be 2.0 since this is the suggested cut off for those 65 and above. Also, I believe there is some suggestion that the fib 4 level for those with diabetes should be at a different level than either 1.3 or 2.0 as the first level of detection of fibrosis. You may want to investigate this and provide an analysis that determines if these cut offs perform better.
In addition, your figures do not seem to make sense to me- especially as you present OR's - where did they come from? I would assume you are only trying to show differences in groups but I do not think the figure displays this well.
Comments on the Quality of English LanguageThere are just a few English grammar mistakes-
Author Response
I think the study can be relevant; however, your results do not match your conclusion- the Fib 4 is not associated with higher risk of CVD or extrahepatic cancer. Fib 4 is not statistically significant for CVD in the multivariate analysis and the results for extrahepatic cancer were missing.
REPLY: We thank the reviewer for their valuable comment. Indeed, FIB-4 was not associated with a higher risk of CVD or extrahepatic cancer. However, T2DM patients with a CVD or cancer history had higher FIB-4 scores at enrollment compared to those patients without a CVD or cancer history. In addition, FIB-4>1.3 was found to be correlated with higher rates of CVD and extrahepatic cancer history. We clarified that better through the whole body of the manuscript and the discussion. Furthermore, we changed the conclusions (page 18, first paragraph) and corrected the conclusion section in the abstract (page 2). Regarding the results of extrahepatic cancer, we added a section on page 13 (page 13, last paragraph and first lines of page 14; "In the univariate analysis, FIB-4>1.3 (OR 2.697, 95%CI 1.311-5.546; p=0.007) and age (OR 1.036, 95%CI 1.000-1.073; p=0.05) at enrollment were associated with a history of extrahepatic cancer. Due to FIB-4's age-related nature, a multivariate analysis was not performed. Including FIB-4 and age in a multivariate analysis would likely result in questionable outcomes due to multicollinearity issues").
Although as you note that age is a component of FIb-4, you removed age from the model and Fib-4 became not significant, this suggests that age may be a significant contributor to the observed relationship. In fact, given the age of the population of study, a better Fib-4 cut off to use may be 2.0 since this is the suggested cut off for those 65 and above.
REPLY: We thank the reviewer for this valuable comment. We added a section about the correlation between FIB-4 and age and the proposed FIB-4 cut-off >2 in patients over 65 (page 4, 2nd paragraph). We also investigated if T2DM patients older than 65 having FIB-4>2 have higher rates of CVD or cancer history (page 14).
Also, I believe there is some suggestion that the fib 4 level for those with diabetes should be at a different level than either 1.3 or 2.0 as the first level of detection of fibrosis. You may want to investigate this and provide an analysis that determines if these cut offs perform better.
REPLY: Several FIB-4 cut-offs have been applied to better evaluate the liver fibrosis status of T2DM patients. For example, Kawata et al. identified that FIB-4>2.96 could be more accurate in detecting cirrhosis (not significant fibrosis) in these patients. Ajmera et al. also reported that a FIB-4 threshold of 1 might increase the test's sensitivity in ruling out significant fibrosis in T2DM patients. We have mentioned these studies on page 4. However, researchers have not yet concluded which should be the optimal FIB-4 cut-off for ruling out significant fibrosis in T2DM patients, and the several proposed cut-offs need further validation. Therefore, the international guidelines have not been changed, and the threshold of 1.3 remains the recommended threshold for ruling out significant fibrosis in the general population and in T2DM patients. Considering the above, we decided to use the FIB-4 threshold of 1.3 in our study. Besides, the study aimed not to use the most sensitive cut-off to exclude significant fibrosis but to investigate whether the threshold recommended by international guidelines of 1.3 could differentiate patients with a history of CVD or cancer from those without a history.
In addition, your figures do not seem to make sense to me- especially as you present OR's - where did they come from? I would assume you are only trying to show differences in groups but I do not think the figure displays this well.
REPLY: The reviewer is correct. The addition of ORs, which came from the results of the univariate analysis, led to confusion. Therefore, we removed that from the figures and mentioned it in the main body of the manuscript (page 13).
Comments on the Quality of English LanguageThere are just a few English grammar mistakes-
REPLY: Editing the English language was performed by a colleague living and working in the UK.
Round 2
Reviewer 1 Report
Comments and Suggestions for Authors
The authors did not answer most of the above questions; therefore, the results obtained are not very impressive, also in view of the retrospective nature of the study.
Comments on the Quality of English LanguageOnly minor corrections are needed
Author Response
The authors did not answer most of the above questions; therefore, the results obtained are not very impressive, also in view of the retrospective nature of the study.
REPLY: We have already answered all of the reviewer's comments. After communicating with the editorial manager, it became clear that the reviewer had not received our answers due to a system problem. Regarding the author's comment that the results of our study are not impressive, we absolutely disagree. The retrospective nature of our study is a limitation, but it does not diminish the value of the results. Prospective studies involving larger cohorts are definitely necessary to investigate whether FIB-4 scores greater than 1.3 could be used as a predictor of CVD and cancer in patients with T2DM. This is clearly mentioned in the discussion.
Herein, you can see our answers to the previous reviewer's comments (round 1):
"As a preliminary step, in order to make their manuscript more readable, authors must provide for a thorough revision, including the tables."
REPLY: We thank the reviewer for their comment. We carefully revised the entire manuscript and tables, correcting grammatical errors, rephrasing some sentences, and more thoughtfully describing our results. Furthermore, we corrected some mistakes in the tables. Hence, no more differences exist between the tables' results and those provided in the manuscript's main body.
"I have some difficulty interpreting some of the results presented, such as the very high risk in women for CVD or the absence of association with lipid profile. The authors need to explain these findings."
REPLY: Regarding the high risk of CVD in women with T2DM, this has already been reported by other investigators as well (references 26,27,28). It has been speculated that women can accumulate more fat and experience a more significant insulin resistance, remaining in a prediabetic state for a more extended period compared to males until the development of T2DM. This prolonged stay in a prediabetic state may be responsible for the higher risk of CVD in female T2DM patients compared to males. We have mentioned that in the discussion (page 16, 2nd paragraph).
Regarding the absence of any association between CVD and the lipidemic profile, we have mentioned that the lipid profile provided is at the time of enrollment. Due to the study's retrospective nature, the HDL, LDL, and Tg values before the onset of CVD are lacking. Unfortunately, patients with a history of CVD had already started taking hypolipidemic medication at the time of the event, and, therefore, they had an improved lipid profile when included in the study. That is why an association between the lipid profile at enrollment and a CVD history was not found. We clarified that in the discussion (page 15, last paragraph, and on page 17, 3rd paragraph, where the study's limitations are highlighted).
"Several evidences have demonstrated the anti-tumor effect of Metformin; it would have been useful for the Authors to correct analysis also for pharmacological treatment."
REPLY: We did not have information about patients' medical treatment during the whole course of T2DM. We had information about antidiabetic therapy at enrollment. We have mentioned that limitation on page 15, last paragraph, and on page 16, 1st paragraph ("A significant difference was observed at enrollment in the number of patients treated with GLP-1 analogs or SGLT-2 inhibitors between the CVD and the non-CVD group. This was mainly because a higher number of patients had started using the mentioned antidiabetic agents after developing CVD in order to benefit from their potential cardioprotective effects. Thus, patients with a history of CVD were already taking these pharmaceutical agents at higher rates at the time of inclusion in the study."). We have also mentioned it on page 17, 3rd paragraph (" the retrospective evaluation of patients did not allow us to investigate the potential effect of the antidiabetic treatment on CVD and the risk of extrahepatic cancer. The antidiabetic treatment at the enrollment was reported, but information about treatment modifications during T2DM and the duration of each administrated treatment is missing".)
Nevertheless, according to the reviewer's proposal, we analyzed whether patients taking Metformin at the time of recruitment had lower rates of extrahepatic cancer history. No significant association was found between metformin treatment and cancer history (χ2 0.386, p=0.551).
"I think the major limitation is represented by the fact that the authors wanted to investigate in the same population the association of FIB-4 with both the risk of cardiovascular events and extrahepatic cancer. The sample size is modest enough to be able to reach certain conclusions."
REPLY: We believe that the reviewer is right. It is difficult to draw certain conclusions as our study's sample size is small. We mentioned that in the discussion (study's limitations, page 17, 3rd paragraph). We also changed the conclusions to clarify that future prospective studies with larger populations are necessary (page 18, 1st paragraph). However, we believe that our study has some strengths which make it valuable enough. First, it was confirmed that T2DM patients with a CVD or cancer history have higher FIB-4 scores compared to those without a CVD or cancer history. More importantly, it is the first time that the FIB-4 value of 1.3 was shown to be correlated with higher rates of CVD and cancer history. We believe that our results will urge investigators to conduct further studies on this issue. If future studies confirm these findings, FIB-4>1.3 could differentiate patients at higher risk of CVD and cancer regardless of age.
"In view of the proliferative effects of insulin in subjects with insulin resistance, it would have been useful to know the insulinemia values."
REPLY: Unfortunately, due to the study's retrospective nature, information about insulin values is lacking. Comments on the Quality of English Language
"Moderate editing of English language is required."
REPLY: Editing the English language was performed by a colleague living and working in the UK.